# Exploring the Relationship between Built Environment and Commuting Mode Choice: Longitudinal Evidence from China

**DOI:** 10.3390/ijerph192114149

**Published:** 2022-10-29

**Authors:** Chaoying Yin, Xiaoquan Wang, Chunfu Shao, Jianxiao Ma

**Affiliations:** 1College of Automobile and Traffic Engineering, Nanjing Forestry University, Nanjing 210037, China; 2College of Civil and Transportation Engineering, Hohai University, Nanjing 210098, China; 3Key Laboratory of Transport Industry of Big Data Application Technologies for Comprehensive Transport, Beijing Jiaotong University, Beijing 100044, China

**Keywords:** built environment, commuting behavior, longitudinal relationships, life-cycle events, China

## Abstract

The literature has offered much evidence regarding associations between the built environment (BE) and commuting behavior. However, most prior studies are conducted based on cross-sectional samples from developed countries, and little is known about the longitudinal link between BE and commuting behavior. Based on two rounds of survey data from China, this study examines relationships of BE with commuting mode choice from both cross-sectional and longitudinal perspectives. The effects of life-cycle events are considered within a unified framework. Results of the longitudinal examination of BE and commuting mode shift largely support the cross-sectional analysis. Specifically, promoting more balanced land use and improving residential density are important for car use reductions and active travel initiatives. Meanwhile, more balanced land use improves the probability of commuting by motorcycle and electric bike, but reduces the probability of commuting by public transit. This study also highlights the remarkable role played by life-cycle events in affecting commuting mode shifts.

## 1. Introduction

Many countries are suffering from urban problems such as air pollution and traffic congestion, which are highly correlated with car dependence [1]. The literature suggests that efforts from two aspects (i.e., urban planning and transport policy) have the potential to reduce car dependence because people may change their travel behavior as a response to built environment (BE) changes and life-cycle events [2,3,4,5]. However, most prior studies on the determinants of changes in travel behavior are conducted in contexts of developed countries [6], and little is known in developing countries. Differences in travel habits and travel choice sets limit the potential to generalize the findings to developing countries. Many typical developing countries (e.g., China) are undergoing urbanization and characterized by city sprawl and changes in social features. These characteristics make them an interesting context for research on the effects of BE changes and life-cycle events on changes in commuting mode choice.

Although some researchers have attempted to examine the factors influencing commuting mode choice, there remain some research gaps to be filled. First, knowledge about BE and commuting mode choice are primarily analyzed based on cross-sectional data [7,8,9], and the longitudinal link between BE and commuting mode choice remains underexplored. Most cross-sectional studies have demonstrated that surrounding BE attributes are determinants of commuting mode choice [10,11]. Cross-sectional studies cannot meet the time precedence [4,7,8], which is a criterion for inferring causality. Thus, it is important to model BE and commuting mode choice based on longitudinal data. Some longitudinal studies have found that people usually change their travel behavior following a residential relocation [12,13,14]. The changes may be primarily explained by the reason that changes in BE and social environment change the choice sets or the utilities of the choices, thus triggering a reconsideration of people’s travel behavior [15]. Therefore, it is expected that BE has a longitudinal relationship with commuting mode choice. In other words, changes in BE may precipitate commuting mode shift. However, it remains largely unclear whether and to what extent BE changes precipitate changes in commuting mode choice. Second, most prior studies have drawn disproportionally on data from developed countries [16]. However, determinants of commuting mode shift in developing countries are relatively unstudied. People from different cultural backgrounds not only show different preferences for different travel modes [17], but also have different choice sets and utilities for each choice. For instance, people tend to have preferences for public transit and active modes in China than those in developed countries. Meanwhile, as electric bikes and motors can provide flexible and convenient travel experience relative to other modes, they are viewed as competitive options to cars in China and Southeast Asian countries [18]. Thus, many local governments encourage people to use electric bikes and motorcycles to meet motorized travel demand because of their smaller contributions to air pollution and traffic congestion. Use of electric bikes and motorcycles is restricted in some cities as they have been criticized owing to safety concerns. In addition, scheduling a child escort is popular in China because of travel safety concerns [19], which is a main factor diving people to purchase a car in many Chinese households. Therefore, it is important to consider these related characteristics when understanding the determinants of commuting mode shifts in a Chinese context.

To fill the above gaps, this study employs logit regression models to examine both cross-sectional and longitudinal relationships of commuting mode choice with BE and life-cycle events using longitudinal survey data from the China Labor-force Dynamics Survey (CLDS). We summarize the contributions of this study as follows:This study provides additional evidence on the relationships between commuting mode choice and BE from a longitudinal perspective. We also perform comparative analysis of the cross-sectional and longitudinal relationships between BE and commuting mode choice.The large sample size and breadth of CLDS promise jointly examining relationships of life-cycle events and BE changes in commuting mode shift.

The remainder of this study begins with a brief review of related studies. Section 3 presents the data and modeling framework of this study. Section 4 and Section 5 present and discuss the estimation results, respectively. The final section summarizes essential findings.

## 2. Literature Review

### 2.1. Life-Cycle Events and Travel Behavior

Most people deliberate their daily travel choices, which tend to be habitual, only when they experience a significant change in their life or environment. Key life-cycle events may disrupt the routine [20], and further induce reconsiderations of daily travel behavior. According to Lanzendorf’s mobility biography framework, researchers should attach importance to life-cycle events in several domains, including lifestyle, accessibility and vehicle ownership [4]. 

Regarding the lifestyle domain, marriage status, childbirth, and changed jobs receive most research interest [21,22,23]. Marriage and divorce may affect motorized travel demand of households [24]. The childbirth in a household can cause more tasks correlated to health and education purposes [25]. Thus, couples may reduce inconvenient ways to take care of children, such as public transit and cycling. People usually reconsider their commuting behavior after changing jobs [2]. Changed jobs may change people′s commuting time and distances, and thus affect their daily travel behavior [6]. Meanwhile, changed jobs may change people′s social networks and further affect activity-travel alternatives [26]. 

Another stream of literature focuses on the accessibility domain (e.g., residential and workplace locations). Many empirical studies have found that people′s travel behavior usually changes after residential relocations [13,27,28]. Residential relocations provide a different context composed of both social environment and BE [6,29]. Thus, people may have different choices available, and the utilities of the choices also change, which can precipitate changes in travel behavior. Additionally, car ownership mediates the links between changes in social environment and BE associated with relocations and travel behavior changes [15]. Residential relocations also affect travel behavior through changing people’s commuting distances, time, and costs. For example, Zhao and Zhang examined the influence mechanism of residential relocations on changes in travel behavior and found that changes in commuting distances precipitate changes in active trips [29].

Associations of vehicle ownership with changes in travel behavior have also received extensive research interest. As vehicle use is, to a large extent, a consequence of vehicle ownership [30], vehicles acquired precipitate a reconsideration of daily travel behavior. Many empirical studies have confirmed that car ownership is a determinant of car use [31,32,33,34]. In other words, people would utilize a car once they own one [35]. Thus, some studies found that buying a car precipitated people to switch from other modes to car [1,5]. Meanwhile, changes in car ownership serve as a mediator between changes in life contexts and travel behavior. Changes in life-cycle events cause changes in car ownership [21]. Changes in car ownership can precipitate changes in travel behavior (e.g., frequency, time, and mode) [15,36]. Acquiring an electric bike can also change daily travel behavior [37,38]. For example, Sun et al. found that acquiring an electric bike largely reduced car use in less urbanized areas based on survey data from Netherlands [38].

### 2.2. BE and Travel Behavior

Prior studies have provided extensive evidence on cross-sectional relationships between BE and travel behavior from different aspects [39,40,41,42,43,44,45,46]. In the literature, residential BE has been measured at different spatial scales, including different buffers [47,48,49], traffic analysis zones [50], communities and census block [51,52]. Recently, spatial scales have been extended from residential locations to workplace locations and city levels [30,32,53,54]. Prior studies have also explored various travel-related outcomes, such as travel mode, duration, frequency, distance, vehicle miles traveled and travel carbon dioxide [55,56,57]. Although prior cross-sectional studies have provided mixed evidence due to various methodologies and contexts, most of them confirm that higher density, greater connectivity and more balanced land use are important for increasing active trips [58]. However, these studies only capture the cross-sectional associations but not longitudinal relationships [5,59]. The shortcoming of cross-sectional associations mainly comes from residential self-selection. For example, do people living in walk-friendly neighborhoods generate more walking trips because the BE causes them to walk more, or do these people choose to live in these neighborhoods because of its walk-friendly environment? As criteria for inferring causality, time precedence is difficult to be met based on cross-sectional data [60]. Thus, it is important to model BE and commuting mode choice based on longitudinal data because it can provide a temporal continuity across individuals. 

Some prior studies have attempted to examine the longitudinal relationships between BE and travel behavior using panel survey data [5,6,15,59,61,62]. Using quasi-longitudinal data, Cao et al. confirmed the remarkable role played by BE in affecting car ownership by combining quantitative and qualitative analysis [61]. Wu and Hong used two round of survey data to analyze subway system expansion and travel behavior and found that improving employment accessibility reduces car use [62]. Based on data from UK Household Longitudinal Study, Clark et al. found that people changed their commuting behavior in response to neighborhood BE changes [6]. However, most prior studies on longitudinal relationships are conducted based on small-scale samples from developed countries [3], which provide limited implications for promoting sustainable transport in developing countries.

### 2.3. Conceptual Framework

Figure 1 presents a unified conceptual framework of relationships among life-cycle events, BE and commuting behavior. The links are explored from both cross-sectional and longitudinal perspectives (solid lines) in this study. Commuting choices are a deliberative process [63]. Regarding the cross-sectional perspective, socio-economics and BE play a remarkable role in shaping commuting behavior, which has been extensively demonstrated in the literature [9,40,41]. From the longitudinal perspective, most people deliberate their daily travel choices, which tend to be habitual, only when they experience a significant change in their life contexts [2]. Among contextual changes, life-cycle events and BE changes are essential triggers. People′s commuting behavior is not deliberated over each time. It is performed until life-cycle events disrupt everyday routines [1,64]. These events may trigger a reconsideration of their commuting behavior, and thus cause commuting mode shift. Apart from life-cycle events, BE changes may lead to changes in commuting behavior as BE variables are typical measurements of accessibility, option availability and diversity. Many studies have confirmed the associations of BE with commuting behavior [10,65,66,67]. It is expected that BE has a longitudinal relationship with commuting behavior. This is reasonable because travel behavior is, to a large extent, a consequence of BE and social environment [15]. Along with contextual changes, the choice sets may change, and the utilities of the choices also change. According to the maximum utility theory, people may change their travel choices when the choice set or the utilities of the choices change. Because a new context is composed of a prior life context and its changes [6], a deliberative process of changing commuting behavior is also affected by life contexts at wave one in the framework.

Overall, it is expected that BE should have a longitudinal relationship with travel behavior, in addition to the cross-sectional association. However, the issue of causality is often assumed in most prior studies, particularly in the contexts of developed countries. The cross-sectional studies cannot address the time precedence in BE and commuting mode choice. Based on large-scale longitudinal survey data from China, this study addresses these gaps by examining relationships between BE and commuting mode choice from both cross-sectional and longitudinal perspectives. The effects of life-cycle events are captured simultaneously in a unified framework.

## 3. Methodology

### 3.1. Data

This analysis is conducted based on two-wave survey data from CLDS 2014 (wave one) and 2016 (wave two). The CLDS is a survey covering 29 provinces in China, conducted by Sun Yat-sen University. The CLDS is a longitudinal survey, which builds a tracking database every two years since 2012. The survey is conducted between July and September in survey years. To make it representative of the nation labor-force population, a multi-stage cluster and stratified probability proportionate to size sampling method is used [68]. This study aims to explore the relationships between BE and commuting mode choice from both cross-sectional and longitudinal perspectives. Thus, only the respondents who participated in two rounds of surveys were selected in this study. The CLDS contains several subsets: individual, household, and neighborhood subsets. The dataset captures a range of fields, including social, economic, and environment information. After data cleaning, records of 6576 respondents from 105 cities are used in this study (Figure 2).

Respondents were asked to report how they got to their workplaces. Accordingly, five modes, including car, active modes, motorcycles/electric bike, public transit and other modes are recorded in the dataset. Four modes, including car, active modes (i.e., walking and cycling), motorcycles/electric bike and public transit, are analyzed in this study. According to Table 1, 3356 (51.04%) respondents commuted by active modes at wave one and 38.01% of active commuters chose other modes at wave two. The percentage of commuters using motorcycles or electric bikes accounted for 23.43% at wave one, which ranked the second place. Additionally, 910 (18.07%) respondents commuting by other modes switched to motorcycle/electric bike. 354 (5.38%) respondents commuted by car at wave one while 401 (6.09%) respondents commuted by public transit.

Statistical descriptions of independent variables are presented in Table 2. The dataset contains life-cycle events and socio–economic information. The following life-cycle events occurring between 2014 and 2016 are collected: (1) changed jobs, (2) marriage, (3) divorce; (4) car acquired; (5) motorcycle/electric bike acquired, and (6) childbirth. All life-cycle events are measured with dummy variables. Four hundred and fifteen (6.31%) and five hundred and thirty-six (8.15%) respondents changed their jobs or experienced a childbirth, respectively. One thousand five hundred and sixty (23.72%) and five hundred and fifteen (7.83%) respondents got married or divorced, respectively. Eight hundred and sixty-two (13.11%) and three thousand one hundred and seventy-eight (48.32%) respondents bought a car or motorcycle/electric bike, respectively. Regarding BE variables and their changes, we only focus the BE around residential locations. The community type and distance to the town government or neighborhood committee are directly extracted from the dataset. The dataset contains population sizes and community areas of neighborhoods, which are used to calculate the population density. Seven categories of facilities (i.e., banks, hospitals, reading rooms, sports facilities, squares, playgrounds, and senior centers) are used to calculate the land use mix based on the entropy approach as follows:(1)$$LUM=\frac{-\sum_{i=1}^{s}p_{i}\ln(p_{i})}{\ln s}$$
where $p_{i}$ represents the ratio of *i*th category of facilities in a neighborhood. *s* represents the number of categories.

### 3.2. Methodology

Three categories of dependent variables are analyzed: commuting by one specific mode in wave one and switching from and to this position. This analysis is twofold by combining cross-sectional and longitudinal examinations. The purpose of the cross-sectional analysis is to explore relationships between BE and socio–economic variables with commuting by one specific mode based on the sample at wave one. The purpose of the longitudinal analysis is to explore relationships of changes in BE and life-cycle events with commuting mode shift based on the sample at both waves. 

Binary logit models are utilized for both cross-sectional and longitudinal models. In cross-sectional models, each mode is represented by a binary variable. It takes a value of ‘1’ for a respondent commuting by one certain mode or ‘0’ otherwise. In longitudinal models, the dependent variable takes a value of ‘1’ when a respondent switches to or from this position. Commuting mode choice or shift is a typical discrete choice problem. People tend to choose the option with the maximum utility in the choice set. The utility function consists of a fixed and random part. In the cross-sectional model, the fixed part consists of socio-economics and neighborhood BE. In the longitudinal model, the fixed part consists of life-cycle events and changes in BE besides socio-economics and neighborhood BE. In logit models, the random part is assumed to follow a Gumbel distribution.

## 4. Results

### 4.1. Commuting by Car and Changes to and from This Position

Table 3 presents the estimation results of commuting by car and changes to and from this position. Regarding BE, all variables show significant effects on car commuting. The negative effect of RD suggests that people in areas with higher RD are less likely to commute by car. This result is consistent with most existing studies [57,69]. LUM shows a negative effect on commuting by car. The positive effect of CT suggests that people in urban areas are more likely to commute by car. DTTN shows a positive effect on car commuting, suggesting that people who live farther from the town government or neighborhood committee are more likely to commute by car. GENDER shows a positive effect on car commuting, suggesting that men are more likely to commute by car. The negative effect of AGE suggests that older people are less likely to commute by car. The positive effects of EDU and HS suggest that well-educated people from bigger households are more likely to commute by car.

Most characteristics show significant effects on switching to and from car. Specifically, GENDER shows a positive and negative effect on switching to and from car respectively. AGE shows positive and negative effects on switching from and to car, respectively. EDU shows positive effects on both switching to and from car. HS shows a positive effect on switching to car. Switching to and from car tend to occur for people who changed jobs between two waves. People who married between two waves show significant modal shift to car, whereas those who divorced between two waves show significant modal shift from car. The uptake of cars and electric bikes shows significantly contrary effects on two switching decisions. The uptake of cars precipitates people to switch to car, whereas the uptake of electric bikes has a negative effect on switching to car. Meanwhile, the uptake of cars has a negative effect on switching from car, whereas the uptake of electric bikes precipitates people to switch from car. CHILD shows a positive effect on switching to car, suggesting that childbirth may precipitate people to switch to car. This is probably because it is more convenient to take the child out by car than other modes [2]. Regarding changes in BE, CRD, LUM, and DTTN show significant effects on modal shift to and from car. The increase in RD has a negative effect on switching to car and precipitates people to switch from car. This finding is consistent with most existing results based on cross-sectional analysis [70], in which researchers found that promoting high-density developing strategies can reduce car use. The increase of LUM has a negative effect on modal shift to car. The positive effect of CDTTN suggests that increases in the distance between residences and the town government or neighborhood committee precipitate a modal shift from car to other modes.

### 4.2. Commuting by Active Modes and Changes to and from This Position

Table 4 presents the estimation results of commuting by active modes and changes to and from this position. All BE variables show significant associations with commuting by active modes. The effects of RD, LUM, and DTTN on active commuting are significantly positive, whereas CT shows a significant negative effect. Additionally, the results suggest that older and well-educated men from bigger households have a higher probability of commuting by active modes.

From the longitudinal perspective, most life-cycle events and BE changes are influential factors. JOB only has a negative effect on switching from active modes. MARRIAGE, MOTOR/EB and CHILD not only have negative effects on switching to active modes, but also affect the decisions to switch to active modes. CAR only precipitates people to switch from active modes. The effects of CRD and CLUM support the results of the above cross-sectional analysis. This result indicates that higher residential density and more balanced land use can encourage active commuting. The effect of CCT is not significant. The negative effect of CDTTN suggests that moving to suburban areas has a negative effect on switching to active modes. The coefficients of GENDER, AGE, and HS indicate that older women from smaller households are more likely to become active commuters. EDU only shows a positive association with switching to active modes.

### 4.3. Commuting by Motorcycle/Electric Bike and Changes to and from This Position

Results of the logit regression analysis for motorcycle or electric bike commuting are shown in Table 5. The coefficients of LUM and DTTN are negative and positive, respectively. The results indicate that the probability of commuting by motorcycle/electric bike is higher in communities with less balanced land use or far from the town government or neighborhood committee. The coefficient of GENDER is significantly positive while the coefficients of EDU and HS are significantly negative. The results indicate that younger men with lower education levels prefer commuting by motorcycle or electric bike.

From the longitudinal perspective, JOB precipitates switching to and from motorcycle/electric bike, whereas MARRIAGE only precipitates switching to motorcycle/electric bike. As expected, MOTOR/EB not only precipitates switching to motorcycle/electric bike, but also has a negative effect on switching from motorcycle/electric bike. CAR has a negative effect on switching to motorcycle/electric bike and positive effect on switching from motorcycle/electric bike. Regarding BE changes, only the effect of CRD is not significant. CLUM not only has a negative effect on switching from motorcycle/electric bike, but also positively affects the decision to switch to motorcycle/electric bike. CCT precipitates switching to motorcycle/electric bike. CDTTN shows negative and positive effects on switching to and from motorcycle/electric bike, respectively. GENDER shows positive and negative effects on switching to and from motorcycle/electric bike, respectively. The effects of AGE and EDU on switching from and to motorcycle/electric bike are significant.

### 4.4. Commuting by Public Transit and Changes to and from This Position

Table 6 presents the determinants of public transit commuting and switching to and from this position. The coefficients of LUM and DTTN are significantly negative, while the coefficient of CT is positive. The effect of RD is not significant. Additionally, the results suggest that older people with lower education levels prefer commuting by public transit.

From the longitudinal perspective, DIVORCE precipitates people to switch to public transit, whereas the effect on switching from public transit is not significant. MARRIAGE and CHILD show positive and negative effects on switching from public transit respectively. Regarding BE, only the effects of CLUM and CDTTN are significant, and their effect signs are the same. Both of them have negative effects on switching to public transit and precipitates switching from public transit. Men tend to have a modal shift from public transit. People from bigger households tend to switch from public transit. The coefficients of AGE and EDU indicate that tendencies of older people with lower education levels becoming transit commuters are significant.

## 5. Discussion

This study analyzes the relationship between BE and commuting behavior from both cross-sectional and longitudinal perspectives. Moreover, it provides supplemental evidence for the relationship in an urbanizing context based on large-scale survey data from China. Key findings and policy insights are discussed in this section.

First, this study provides a longitudinal analysis of BE and commuting behavior. The findings indicate that most changes in BE show significant associations with commuting mode shift. Moreover, the longitudinal evidence largely supports the cross-sectional results. Aiming to reduce car use and increase active travel, many planning initiatives based on cross-sectional findings emphasize the benefits of promoting balanced land use [32]. The results show that increases in land use mix not only precipitate switching to active commuting, but also have a negative effect on switching from active commuting and to car commuting. These findings demonstrate the effectiveness of these planning strategies. Additionally, urban sprawl motivated by urbanization has a negative effect on sustainable development because people tend to commute by car, but have a lower probability of commuting by public transit in suburbs. To encourage public transit use in suburbs, ensuring timely provision of public transit is of great importance, as previously highlighted by many cross-sectional studies [11].

Second, this study supplies some evidence of a relationship between the BE and motorcycles/electric bikes in the Chinese context. As competitive options to cars, motorcycles and electric bikes are widely used in developing countries such as China and various Southeast Asian countries [18]. However, the literature has paid limited attention to their relationships with BE. The results show that the land use mix and distance to the town government or neighborhood committee are determinants of motorcycle/electric bike commuting from both cross-sectional and longitudinal perspectives. Due to the benefits of using motorcycles and electric bikes (e.g., lower costs and flexible travel experiences), governments also make efforts to promote motorcycle/electric bike use, such as not requiring a driver’s license. According to the results, providing more balanced land use may be an effective way of promoting motorcycle/electric bike use. Meanwhile, the results show that people in suburbs are more likely to use motorcycles and electric bikes. Thus, policymakers should pay more attention to promoting and managing motorcycle/electric bike use in such areas because the use of motorcycles and electric bike may cause safety concerns.

Third, all life-cycle events have significant effects on commuting mode shift. In particularly, the results show that all life-cycle events, including changed jobs, marriage, divorce, car acquired, electric bike acquired, and childbirth, are determinants of commuting mode shift. These links call for transport policies targeted at different social groups to reduce car use and increase sustainable travel demand, in addition to BE planning and infrastructure construction. For example, transport policies, such as subsidizing policies for motorcycles and electric bikes, may be important for avoiding excessive use of cars for young couples, because marriage and childbirth can increase motorized travel demand. Subsidizing policies for attractive alternatives can be effective because an earlier study suggests that informing young people about the availability of public transit causes more sustainable travel [71].

## 6. Conclusions

This study provides new insights into the effects of the BE variables that influence commuting mode choice based on a large-scale survey dataset from China. It contributes to the literature by providing an analysis of the relationship between BE and commuting mode choice from both cross-sectional and longitudinal perspectives within a unified framework. The findings indicate that both life-cycle events and changes in BE play remarkable roles in affecting commuting mode shift. The effects of changes in BE suggest that promoting more balanced land use and improving residential density are important for reducing car use and encouraging active travel. Additionally, increases in land use mix precipitate the use of motorcycles and electric bikes, but reduce the use of public transit. Changes in distance to the town government or neighborhood committee are associated with most commuting mode shifts, whereas changes in community types are only associated with switching to motorcycle/electric bike. These findings largely support the cross-sectional results. Apart from changes in BE, most life events also have significant effects on commuting mode shift. The findings may not transfer directly to other countries owing to differences in land use and transport, and societal and institutional factors. However, it has important policy implications for other metropolitan regions in China.

Several limitations should be addressed. First, this study only captures direct relationships among the BE, life-cycle events and commuting mode shifts. Future work should consider direct and indirect effects to capture the complex influence mechanism among them in a unified framework. Second, due to data unavailability, preferences for travel and BE are not considered. Preferences for travel and BE are the primary sources of residential self-selection because people’s choice of travel mode may be caused by their preferences but not the BE. Future work should address self-selection effects when exploring relationships. Third, this study only focuses on commuting mode choice. However, the neighborhood BE should be associated with other aspects of travel behavior. Thus, the associations between BE and other aspects of travel behavior need to be explored in future work.

## Figures and Tables

**Figure 1 ijerph-19-14149-f001:**
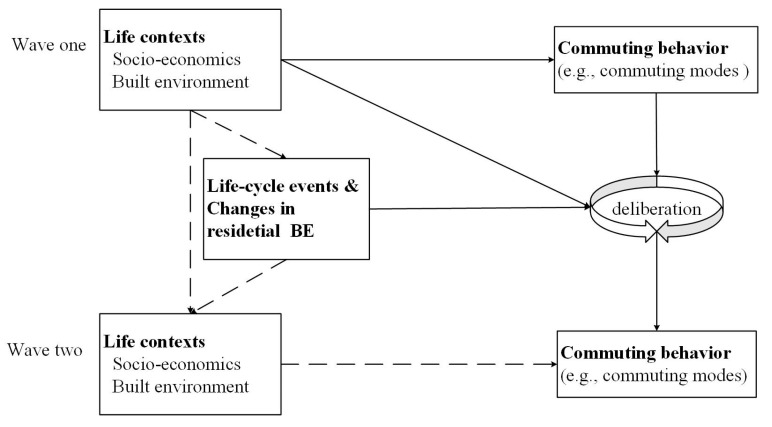
Conceptual framework of this analysis.

**Figure 2 ijerph-19-14149-f002:**
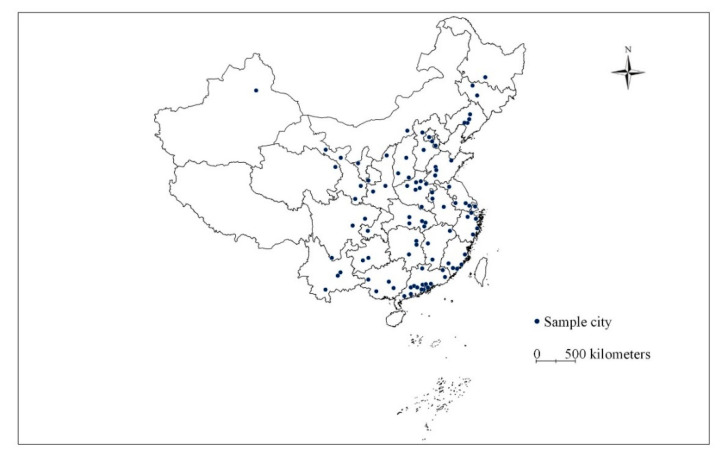
Distribution of the sample.

**Table 1 ijerph-19-14149-t001:** Commuting modes and their changes between two waves.

Variables	Cases	Percentage (%)
Commuting by car at wave one	354	5.38
Switching from commuting by car to other modes	201	56.78
Commuting by active modes at wave one	3356	51.04
Switching from commuting by active modes to other modes	1276	38.01
Commuting by motorcycle/electric bike at wave one	1541	23.43
Switching from commuting by motorcycle/electric bike to other modes	581	37.69
Commuting by public transit at wave one	401	6.09
Switching from commuting by public transit to other modes	226	56.26

**Table 2 ijerph-19-14149-t002:** Variables used for this analysis.

Variables	Description	Mean/Cases	S.E./Percentage
**Life-cycle events**			
JOB	1 = the respondent changed his or her job between two waves; 0 = otherwise	415	6.31
MARRIAGE	1 = the respondent got married between two waves; 0 = otherwise	1560	23.72
DIVORCE	1 = the respondent got divorced between two waves; 0 = otherwise	515	7.83
CAR	1 = the respondent bought a car between two waves; 0 = otherwise	862	13.11
MOTOR/EB	1 = the respondent bought a motorcycle or electric bike between two waves; 0 = otherwise	3178	48.32
CHILD	1 = the respondent experienced a childbirth between two waves; 0 = otherwise	536	8.15
**Change in neighborhood built environment**			
CRD	Residential density at wave two minus its value at wave one	0.87	2.98
CLUM	Land use mix at wave two minus its value at wave one	0.03	0.21
CCT	1 = the community the respondent living in changed from a rural community to an urban community	33	0.72
CDTTN	Distance to the town government or neighborhood committee at wave two minus its value at wave one	0.15	1.35
**Neighborhood built environment (wave one)**			
RD	Population divided by community area (10,000 persons/km^2^)	1.27	0.89
LUM	A measurement of land use diversity using seven types of facilities	0.59	0.24
CT	1 = urban community; 0 = rural community	0.36	0.48
DTTN	Distance from home to the town government or neighborhood committee (km)	3.97	5.42
**Socio-economics (wave one)**			
GENDER	1 = male; 0 = otherwise	3052	46.42
AGE	Age in years	47.86	12.82
EDU	1 = college diploma or above; 0 = otherwise	775	11.78
HS	Number of household members	4.58	1.74

**Table 3 ijerph-19-14149-t003:** Logistic models of car commuting and switches to and from this position.

	Commute by Car (Wave One)		To Car		From Car	
	Coef.	S.E.	Coef.	S.E.	Coef.	S.E.
**Life-cycle events**						
JOB	—	—	0.139 *	0.078	0.319 **	0.108
MARRIAGE	—	—	0.368 **	0.071	0.057	0.049
DIVORCE	—	—	0.347	0.288	−0.105 **	0.040
CAR	—	—	0.267 **	0.047	−0.842 **	0.191
MOTOR/EB	—	—	−0.082 *	0.046	0.175 **	0.033
CHILD	—	—	0.102 **	0.041	0.089	0.094
**Change in neighborhood built environment**						
CRD	—	—	−0.001 **	0.000	0.265 **	0.101
CLUM	—	—	−0.024 **	0.009	0.095	0.066
CCT	—	—	0.049	0.077	0.429	0.299
CDTTN	—	—	0.038	0.039	−0.281 **	0.021
**Neighborhood built environment**						
RD	−0.011 **	0.007	—	—	—	—
LUM	−0.089 *	0.052	—	—	—	—
CT	0.132 **	0.029	—	—	—	—
DTTN	0.028 **	0.013	—	—	—	—
**Socio-economics**						
GENDER	0.196 **	0.278	0.394 **	0.186	−0.563 **	0.389
AGE	−0.292 **	0.147	−0.049 **	0.007	0.261 **	0.007
EDU	0.049 **	0.004	0.352 **	0.187	0.916 **	0.213
HS	0.103 **	0.028	0.001 **	0.050	−0.247	0.151
Constant	0.005 **	0.002	−0.733 **	0.119	0.008 **	0.004
Log likelihood	−869.879		−693.461		−584.666	
Chi squared	701.56		291.29		127.94	
Observations	6576		6222		354	

* Significance level of 10%. ** Significance level of 5%.

**Table 4 ijerph-19-14149-t004:** Logistic models of active commuting and switches to and from this position.

	Commute by Active Modes (Wave One)		To Active Modes		From Active Modes	
	Coef.	S.E.	Coef.	S.E.	Coef.	S.E.
**Life-cycle events**						
JOB	—	—	0.114	0.032	−0.098 **	0.018
MARRIAGE	—	—	−0.109 **	0.031	0.114 **	0.012
DIVORCE	—	—	−0.095	0.128	−0.045	0.050
CAR	—	—	−0.203 **	0.035	0.169	0.187
MOTOR/EB	—	—	−0.122 *	0.040	0.097 **	0.042
CHILD	—	—	−0.253 **	0.104	0.106 **	0.038
**Change in neighborhood built environment**						
CRD	—	—	0.023 **	0.010	−0.106 **	0.021
CLUM	—	—	0.098 **	0.012	−0.293 **	0.074
CCT	—	—	−0.103	0.086	0.096	0.124
CDTTN	—	—	−0.046 **	0.013	0.156	0.201
**Neighborhood built environment**						
RD	0.002 **	0.001	—	—	—	—
LUM	0.013 *	0.008	—	—	—	—
CT	−0.017 **	0.003	—	—	—	—
DTTN	−0.009 **	0.003	—	—	—	—
**Socio-economics**						
GENDER	−0.235 **	0.112	−0.154 **	0.066	0.093 **	0.031
AGE	0.187 **	0.031	0.089 **	0.014	−0.160 **	0.074
EDU	−0.143 **	0.041	0.257 **	0.108	0.106	0.113
HS	−0.079	0.123	−0.011 **	0.043	0.132 *	0.078
Constant	0.019 **	0.010	0.163 **	0.065	0.085 **	0.024
Log likelihood	−1016.185		−776.038		−735.584	
Chi squared	635.13		304.58		185.23	
Observations	6576		3220		3356	

*** Significance level of 10%. ** Significance level of 5%.

**Table 5 ijerph-19-14149-t005:** Logistic models of motorcycle/electric bike commuting and switches to and from this position.

	Commute by Motorcycle/Electric Bike (Wave One)		To Motorcycle/Electric Bike		From Motorcycle/Electric Bike	
	Coef.	S.E.	Coef.	S.E.	Coef.	S.E.
**Life-cycle events**						
JOB	—	—	0.014 **	0.005	0.087 **	0.008
MARRIAGE	—	—	0.114 *	0.060	−0.056	0.048
DIVORCE	—	—	0.097	0.118	0.103	0.240
CAR	—	—	−0.154 **	0.062	0.056 **	0.019
MOTOR/EB	—	—	0.112 *	0.035	−0.089 **	0.012
CHILD	—	—	0.119	0.148	0.075	0.184
**Change in neighborhood built environment**						
CRD	—	—	−0.021	0.101	0.120	0.089
CLUM	—	—	0.101 **	0.031	−0.088 **	0.016
CCT	—	—	0.049 *	0.029	0.065	0.059
CDTTN	—	—	0.048 **	0.015	−0.087 **	0.041
**Neighborhood built environment**						
RD	−0.019	0.022	—	—	—	—
LUM	−0.026*	0.014	—	—	—	—
CT	−0.010	0.018	—	—	—	—
DTTN	0.012 **	0.003	—	—	—	—
**Socio-economics**						
GENDER	0.063 **	0.027	0.109 **	0.036	−0.263 **	0.089
AGE	−0.115 **	0.048	−0.032 **	0.012	0.095 **	0.023
EDU	−0.074 **	0.011	−0.112 **	0.017	0.109 **	0.009
HS	0.089	0.147	0.012	0.045	−0.120	0.141
Constant	0.107 **	0.028	0.156 **	0.069	0.019 **	0.005
Log likelihood	−1356.152		−643.461		−584.667	
Chi squared	569.12		198.35		108.56	
Observations	6576		5035		1541	

* Significance level of 10%. ** Significance level of 5%.

**Table 6 ijerph-19-14149-t006:** Logistic models of public transit commuting and switches to and from this position.

	Commute by Public Transit (Wave One)		To Public Transit		From Public Transit	
	Coef.	S.E.	Coef.	S.E.	Coef.	S.E.
**Life-cycle events**						
JOB	—	—	0.016	0.041	0.113 **	0.038
MARRIAGE	—	—	0.106	0.178	0.019	0.016
DIVORCE	—	—	0.085 **	0.021	−0.081	0.108
CAR	—	—	−0.156 *	0.082	0.062 **	0.011
MOTOR/EB	—	—	−0.097 **	0.026	0.106 **	0.012
CHILD	—	—	0.052	0.048	−0.073 **	0.024
**Change in neighborhood built environment**						
CRD	—	—	−0.021	0.018	0.105	0.101
CLUM	—	—	−0.138 **	0.060	0.105 *	0.056
CCT	—	—	0.027	0.045	0.041	0.039
CDTTN	—	—	−0.072 **	0.029	0.103 **	0.035
**Neighborhood built environment**						
RD	−0.009	0.012	—	—	—	—
LUM	−0.102 **	0.039	—	—	—	—
CT	0.095 *	0.056	—	—	—	—
DTTN	−0.085 **	0.025	—	—	—	—
**Socio-economics**						
GENDER	−0.157	0.205	−0.196 **	0.076	−0.156	0.250
AGE	0.097 **	0.025	0.075 **	0.024	−0.121 **	0.026
EDU	−0.114 **	0.053	−0.102 **	0.014	0.116 **	0.028
HS	0.053	0.045	0.018	0.079	0.148 **	0.038
Constant	−0.109 **	0.028	0.481 **	0.107	0.328 **	0.136
Log likelihood	−1325.558		−523.185		−489.022	
Chi squared	456.81		169.74		89.58	
Observations	6576		6175		401	

* Significance level of 10%. ** Significance level of 5%.

## Data Availability

Not applicable.

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
