# Peer review of "Exploring the Relationship between Built Environment and Commuting Mode Choice: Longitudinal Evidence from China"

_ijerph, 2022, doi:10.3390/ijerph192114149_

Round 1

Reviewer 1 Report

This paper presents new evidence on the causal effects of the built environment on travel behavior, in this case, commute mode. It uses data from China and thus provides much needed evidence from a context outside of North America and Europe. The analysis appears sound. The paper could be improved in several respects, however.

First, given the focus on causality, the authors should take some time to explain causality and the criteria by which we decide that an observed association is in fact a causal relationship. The authors draw a false distinction between cross-sectional and longitudinal analysis, in that cross-sectional analysis is the first step towards establishing causality. The need for controlling for co-variates is another important part of establishing causality.  For reasons I explain below, I suggest that the authors talk about cross-sectional and longitudinal studies, and do not call the latter “causal” studies.

Second, the conceptual model should come earlier in the paper; the end of the lit review section is a good place for this. The authors should have a clear explanation for how the explanatory variables would in fact influence commute mode.

Third, the lack of data on attitudes and preferences needs more attention, along with some discussion of the self-selection issue, which is critical in any discussion about causality.  

Third, the authors need to provide more information about the data collection effort, as noted below.

Finally, the paper needs a thorough editing for awkward wording, e.g. “These characteristics make them become a distinguished research context for…” should be “These characteristics make them an interesting context for research on…” And other examples below, and many more beyond that in the paper.

Specific comments:

Abstract:  “facilitates dependence on motors and electric-bikes, but re-20 duces dependence on public transit.” – What do you mean by dependence? Does “motors” mean motor vehicles?

Pg. 2, lines 47-49: Provide a better explanation of the causal mechanism here. It’s not just that people reconsider their choice but that they have different choices available to them after a move. The choice set may change, and the utilities of the choices also change (according to the usual travel behavior theory).  I’m looking for the kind of explanation you provide starting in line 102.

Pg.2, lines 55-57: Is this simply a matter of different preferences? Again, explain different choice sets and different utilities for each choice (depending on the choice characteristics but also preferences, i.e. the coefficients in the utility equation).

Pg 2, line 69: I would say “cross-sectional and longitudinal relationships” as even a longitudinal analysis is not a perfect test of causality. Similarly, “longitudinal perspectives” in line 73.

Pg 2, line 73: I’m not sure what “their” as in “their relationships” refers to.

Pg. 2, line 84: Restate this in plan English: “Most people deliberate their daily travel choices, which tend to be habitual, only when they experience a significant change in their life or environment.”

Pg. 3, line 110:  By “mobility domain” you apparently mean car ownership, or perhaps vehicle ownership more generally. If so, use these terms instead.

Pg. 3, lines 124-126. The introductory clause does not make sense with the rest of the sentence. The motivation for these studies is to inform such initiatives, not to themselves to directly reduce car dependence.

Pg. 3, line 127: The “5Ds” is not a theoretically sound way to characterize the built environment.

Pg. 3, line 137: Explain why, in this case, the associations may not be a good indicator of causality. In other words, explain residential self-selection.

Pg. 3, lines 140-143: Explain why longitudinal data is important for establishing causality. Explain the criteria for establishing causality!

Pg. 4, line 153:  No, the issue of causality is not overlooked in cross-sectional studies. It is often assumed in cross-sectional studies, and that’s a problem. But the cross-sectional studies are done as a first step towards establishing causality. Explain the criteria for establishing causality!

Pg. 4, line 160:  Who conducts this survey? The national government? Not clear at this point whether this is a panel survey or a repeat cross-sectional survey.  There’s a big difference.  How is the survey conducted? What is the response rate? What is the drop-out rate? What is the replenishment process?

Table 1: What do the two right-hand columns represent? The 201 cases for “commuting by car” – is this the number of people commuting by car in Wave 2 or the number who switched to commuting by car, or switching from commuting by car? What is the percentage – the percentage of wave 1 people who switched? Too many decimal places.  Your “non” modes are confusing (e.g. “non-electric-bike” sounds like a regular bike to me), and you don’t need them anyway – just list the modes and their percentages.

Pg. 5, line 190: This is really important. Why these variables? How and why do you expect them to affect commute mode choice?

Pg. 6: Aha! Here’s the conceptual framework. This should come before you talk about data.  The conceptual model should come out of the literature review and should be presented there. Of course, you already said most of this.

Pg. 6, lines 205-206: “From the causal perspective commuting behavior is a consequence of a deliberative process as commuting behavior is habitual.” If it is habitual, how can it be deliberative? Reword to make your meaning clear.

Results section: Saying that a variable “facilitates” or “deters” a switch is going farther than you should in interpreting results. To say that divorce facilitates a switch to public transit, for example, suggests that this is what people want to do, which I doubt is the case (“precipitates” is probably a more accurate way to describe it). Be careful about your wording. Technically, what you’ve established is an association between a change in an explanatory variable and a change in the dependent variable.

Pg. 12, line 358: How would these results inform policies targeted at different social groups? What would that look like?

Pg. 12, line 368: Not having data on attitudes and preferences is a significant limitation. Rather than ignoring this, the authors need to discuss this limitation with respect to the issue of potential self-selection. And explain what self-selection is.

Pg. 12, line 369: Also, what about the focus on only commute mode? What other aspects of travel behavior would be important to study?

Pg. 12, line 374: “provide appropriate implications” does not make sense. Studies do not “provide” implications. And what are “appropriate implications”?

Section 6 is a summary of findings, not really a conclusion.

Author Response

- Reviewer 1

This paper presents new evidence on the causal effects of the built environment on travel behavior, in this case, commute mode. It uses data from China and thus provides much needed evidence from a context outside of North America and Europe. The analysis appears sound. The paper could be improved in several respects, however.

[Response] Thank you for your comments. We have revised the related contents according to the requirements. We are hoping that the revised manuscript has properly responded to all your comments.

Major comments:

1.1 First, given the focus on causality, the authors should take some time to explain causality and the criteria by which we decide that an observed association is in fact a causal relationship. The authors draw a false distinction between cross-sectional and longitudinal analysis, in that cross-sectional analysis is the first step towards establishing causality. The need for controlling for co-variates is another important part of establishing causality.  For reasons I explain below, I suggest that the authors talk about cross-sectional and longitudinal studies, and do not call the latter “causal” studies.

[Response] Thank for your comments. After thinking twice, we actually explored the association from a longitudinal perspective in this study, but not the causality. Thus, in order to avoid ambiguous expressions, we have revised related words, such as “causal”, “cause”, “lead to”, “facilitate” and “deter” throughout the manuscript. In the new version, we have clearly claimed that we contribute to the literature by investigating the relationships of built environment with commuting mode choice from a longitudinal perspective.

1.2 Second, the conceptual model should come earlier in the paper; the end of the lit review section is a good place for this. The authors should have a clear explanation for how the explanatory variables would in fact influence commute mode.

[Response] Thank for your comments. As you suggest, we have moved the Conceptual framework into the section of literature review.

In order to give a clearer explanation for how the explanatory variables influence commute mode, we have also supplied explanations in the new version. The influence of life events mainly be explained by the reason that life-cycle events disrupt everyday routines and they may trigger a reconsideration of commuting mode. In the manuscript, we have supplied related explanations as follows (lines 161-164): People’s commuting behavior is not deliberated over each time. It is performed until life-cycle events disrupt everyday routines [1, 64, 65]. These events may trigger a reconsideration of their commuting behavior, and thus cause commuting mode shift. Meanwhile, life events and BE changes may influence commute mode because they also affect the choice set or the utilities of choices. We have added explanations as follows (lines 169-172): Along with contextual changes, the choice sets may change, and the utilities of the choices also change. According to the maximum utility theory, people may change their travel choices when the choice set or the utilities of the choices change.

1.3 Third, the lack of data on attitudes and preferences needs more attention, along with some discussion of the self-selection issue, which is critical in any discussion about causality. 

[Response] Thank for your comments. The self-selection issue is actually an important issue in BE and travel behavior. As you suggest, we have supplied discussion about the residential self-selection issue in the section of literature review as follows (lines 133-138): However, these studies only capture the cross-sectional associations but not longitudinal relationships [5, 59]. The shortcoming of cross-sectional associations mainly comes from residential self-selection. For example, do people living in walk-friendly neighborhoods generate more walking trips because the BE causes them to walk more, or do these people choose to live in these neighborhoods because of its walk-friendly environment?

Because we cannot obtain the data on attitudes and preferences, the self-selection issue cannot be addressed. As you suggest, we have supplied it as a limitation as follows (lines 407-411): Second, due to data unavailability, preferences for travel and BE are not considered. Preferences for travel and BE are the primary sources of residential self-selection be-cause people choose a travel mode may be caused by their preferences but not the BE. Future work should address the self-selection effects when exploring the relationships.

1.4 The authors need to provide more information about the data collection effort, as noted below.

[Response] Thank you for comments. We are sorry for the unclear explanations about the data source. The survey data is from an open dataset. Everyone can obtain and use it for research after applying. We have provided the applying website in the manuscript. As you suggest, we have supplied detailed information about the survey data as follows (lines 188-200): This analysis is conducted based on two-wave survey data from CLDS 2014 (wave one) and 2016 (wave two). The CLDS is a survey covering 29 provinces in China, con-ducted by Sun Yat-sen University. The CLDS is a longitudinal survey, which builds a tracking database every two years since 2012. The survey is conducted between July and September in survey years. To make it representative of the nation labor-force population, a multi-stage cluster and stratified probability proportionate to size sampling method is used (http://css.sysu.edu.cn for detail) [63]. This study aims to explore the relationships between BE and commuting mode choice from both cross-sectional and longitudinal perspectives. Thus, only the respondents who participated in two rounds of surveys were selected in this study. The CLDS contains several subsets: individual, household, and neighborhood subsets. The dataset captures a range of fields, including social, economic, and environment information. After data cleaning, records of 6576 respondents from 105 cities are used in this study (Figure 2).

1.5 Finally, the paper needs a thorough editing for awkward wording, e.g. “These characteristics make them become a distinguished research context for…” should be “These characteristics make them an interesting context for research on…” And other examples below, and many more beyond that in the paper.

[Response] Thank for your comments. We have revised related contents. Also, we have invited a professional, who has published papers on many international journals, to help us proof our paper. Thank you for your efforts on improving our paper.

Minor comments:

1.6 Abstract:  “facilitates dependence on motors and electric-bikes, but re-20 duces dependence on public transit.” – What do you mean by dependence? Does “motors” mean motor vehicles?

[Response] We feel sorry for the unclear expressions. The motors mean motorcycles. And we have revised this sentence as follows (lines 19-21): Meanwhile, more balanced land use improves the probability of commuting by motorcycles and electric-bikes, but reduces the probability of commuting by public transit.

1.7 Pg. 2, lines 47-49: Provide a better explanation of the causal mechanism here. It’s not just that people reconsider their choice but that they have different choices available to them after a move. The choice set may change, and the utilities of the choices also change (according to the usual travel behavior theory).  I’m looking for the kind of explanation you provide starting in line 102.

[Response] Thank you for your comments. As you suggest, we have tried to add a better explanation of the causal mechanism as follows (lines 48-50): The changes may be primarily explained by the reason that changes in BE and social environment change the choice sets or the utilities of the choices, thus triggering a reconsideration of people’s travel behavior [15].

As you suggest, we also add explanations in the section of literature review as follows (lines 102-105): Residential relocations provide a different context composed of both social environment and BE [6, 29]. Thus, people may have different choices available and the utilities of the choices also change, which can precipitate changes in travel behavior.

1.8 Pg.2, lines 55-57: Is this simply a matter of different preferences? Again, explain different choice sets and different utilities for each choice (depending on the choice characteristics but also preferences, i.e. the coefficients in the utility equation).

[Response] Thank you for your comments. We have supplied expressions about different choice sets and different utilities for each choice besides preferences as follows (lines 56-57): People from different cultural backgrounds not only show different preferences for different travel modes, but also have different choice sets and utilities for each choice.

We have also supplied an example for the explanations about different choice sets in China and Western countries as follows (lines 59-61): Meanwhile, as electric-bikes and motors can provide flexible and convenient travel experience relative to other modes, they are viewed as competitive options to cars in China and Southeast Asian countries [18].

1.9 Pg 2, line 69: I would say “cross-sectional and longitudinal relationships” as even a longitudinal analysis is not a perfect test of causality. Similarly, “longitudinal perspectives” in line 73.

[Response] Thank you for your comments. As you suggest, we have revised the causal into longitudinal in the new version. Besides the sentence you provide, we revised all related contents.

1.10 Pg 2, line 73: I’m not sure what “their” as in “their relationships” refers to.

[Response] Thank you for your comments. To make it clearer, we have revised “their relationships” into “the cross-sectional and longitudinal relationships between BE and commuting mode choice” in the new version.

1.11 Pg. 2, line 84: Restate this in plan English: “Most people deliberate their daily travel choices, which tend to be habitual, only when they experience a significant change in their life or environment.”

[Response] Thank you for your comments for improving our paper. We have replaced the sentence with the sentence you provid.

1.12 Pg. 3, line 110:  By “mobility domain” you apparently mean car ownership, or perhaps vehicle ownership more generally. If so, use these terms instead.

[Response] Thank you for comments. Vehicle ownership can represent the variable more accurately. And we have replaced mobility domain” with “vehicle ownership” in the new version.

1.13 Pg. 3, lines 124-126. The introductory clause does not make sense with the rest of the sentence. The motivation for these studies is to inform such initiatives, not to themselves to directly reduce car dependence.

[Response] Thank you for comments. As you suggest, we have deleted the introductory clause of the sentence.

1.14 Pg. 3, line 127: The “5Ds” is not a theoretically sound way to characterize the built environment.

[Response] Thank you for comments. As you suggest, we have deleted related contents.

1.15 Pg. 3, line 137: Explain why, in this case, the associations may not be a good indicator of causality. In other words, explain residential self-selection.

[Response] Thank you for comments. As you suggest, we have supplied several sentences to explain why the associations may not be a good indicator of causality as follows (lines 133-138): However, these studies only capture the cross-sectional associations but not longitudinal relationships [5, 59]. The shortcoming of cross-sectional associations mainly comes from residential self-selection. For example, do people living in walk-friendly neighborhoods generate more walking trips because the BE causes them to walk more, or do these people choose to live in these neighborhoods because of its walk-friendly environment?

1.16 Pg. 3, lines 140-143: Explain why longitudinal data is important for establishing causality. Explain the criteria for establishing causality!

[Response] Thank you for comments. As you suggest, we have supplied several sentences to explain why longitudinal data is important for establishing causality (lines 138-141): As criteria for inferring causality, time precedence is difficult to be met based on cross-sectional data [60]. Thus, it is important to model BE and commuting mode choice based on longitudinal data because it can provide a temporal continuity across individuals.

1.17 Pg. 4, line 153:  No, the issue of causality is not overlooked in cross-sectional studies. It is often assumed in cross-sectional studies, and that’s a problem. But the cross-sectional studies are done as a first step towards establishing causality. Explain the criteria for establishing causality!

[Response] Thank you for comments. As you suggest, we have revised the sentence as follows (lines 179-182): However, the issue of causality is often assumed in most prior studies, particularly in the contexts of developed countries. The cross-sectional studies cannot address the time precedence in BE and commuting mode choice.

1.18 Pg. 4, line 160:  Who conducts this survey? The national government? Not clear at this point whether this is a panel survey or a repeat cross-sectional survey.  There’s a big difference.  How is the survey conducted? What is the response rate? What is the drop-out rate? What is the replenishment process?

[Response] Thank you for comments. As you suggest, we have supplied detailed information about the survey data as follows (lines 188-200): This analysis is conducted based on two-wave survey data from CLDS 2014 (wave one) and 2016 (wave two). The CLDS is a survey covering 29 provinces in China, con-ducted by Sun Yat-sen University. The CLDS is a longitudinal survey, which builds a tracking database every two years since 2012. The survey is conducted between July and September in survey years. To make it representative of the nation labor-force population, a multi-stage cluster and stratified probability proportionate to size sam-pling method is used (http://css.sysu.edu.cn for detail) [63]. This study aims to explore the relationships between BE and commuting mode choice from both cross-sectional and longitudinal perspectives. Thus, only the respondents who participated in two rounds of surveys were selected in this study. The CLDS contains several subsets: indi-vidual, household, and neighborhood subsets. The dataset captures a range of fields, including social, economic, and environment information. After data cleaning, records of 6576 respondents from 105 cities are used in this study (Figure 2).

1.19 Table 1: What do the two right-hand columns represent? The 201 cases for “commuting by car” – is this the number of people commuting by car in Wave 2 or the number who switched to commuting by car, or switching from commuting by car? What is the percentage – the percentage of wave 1 people who switched? Too many decimal places.  Your “non” modes are confusing (e.g. “non-electric-bike” sounds like a regular bike to me), and you don’t need them anyway – just list the modes and their percentages.

[Response] Thank you for comments. We are sorry for the unclear expressions in last version. We have revised the table as follows:

Variable

Cases

Percentage (%)

Commuting by car at wave one

354

5.38

Switching from commuting by car

201

3.06

Commuting by active modes at wave one

3356

51.04

Switching from commuting by active modes

1276

19.40

Commuting by motor/electric-bike at wave one

1541

23.43

Switching from commuting by motor/electric-bike

581

8.84

Commuting by public transit at wave one

401

6.09

Switching from commuting by public transit

226

3.44

1.20 Pg. 5, line 190: This is really important. Why these variables? How and why do you expect them to affect commute mode choice?

[Response] Thank you for comments. We choose these variables because of the following reasons. These variables are common facilities used to measure the built environment, which have been popularly used in the literature. And previous studies use more than or less than seven facilities to calculate the land use mix. As long as the facilities can reflect the land use, the number of types of facilities is mainly decided by the data available. However, we cannot obtain more facility information from the databased because the neighborhood names are anonymous to protect privacy. Of course, more facilities are better because they can better represent land use mix if more facilities information can be obtained.

1.21 Aha! Here’s the conceptual framework. This should come before you talk about data.  The conceptual model should come out of the literature review and should be presented there. Of course, you already said most of this.

[Response] Thank you for comments. As you suggest, we have moved this figure into the Section of Literature Review.

1.22 Pg. 6, lines 205-206: “From the causal perspective commuting behavior is a consequence of a deliberative process as commuting behavior is habitual.” If it is habitual, how can it be deliberative? Reword to make your meaning clear.

[Response] Thank you for comments. As you suggest, we have revised the sentence as follows (lines 158-160): From the longitudinal perspective, most people deliberate their daily travel choices, which tend to be habitual, only when they experience a significant change in their life contexts [2].

1.23 Results section: Saying that a variable “facilitates” or “deters” a switch is going farther than you should in interpreting results. To say that divorce facilitates a switch to public transit, for example, suggests that this is what people want to do, which I doubt is the case (“precipitates” is probably a more accurate way to describe it). Be careful about your wording. Technically, what you’ve established is an association between a change in an explanatory variable and a change in the dependent variable.

[Response] Thank for your comments. In order to avoid ambiguous expressions, we have revised related words, such as “causal”, “cause”, “lead to”, “facilitate” and “deter” throughout the manuscript.

1.24 Pg. 12, line 358: How would these results inform policies targeted at different social groups? What would that look like?

[Response] Thank you for comments. As you suggest, we have supplied an example for potential transport policies as follows (lines 379-384): For example, transport policies, such as subsidizing policies for motorcycles and electric-bikes, may be important to avoid excessive use of cars for young couples because marriage and childbirth can increase motorized travel demand. The subsidizing policies for attractive alternatives can be effective because an earlier study suggests that informing young people about the availability of public transit causes more sustainable travel [71].

1.25 Pg. 12, line 368: Not having data on attitudes and preferences is a significant limitation. Rather than ignoring this, the authors need to discuss this limitation with respect to the issue of potential self-selection. And explain what self-selection is.

[Response] Thank you for comments. As you suggest, we have revised the limitation as follows (lines 407-411): Second, due to data unavailability, preferences for travel and BE are not considered. Preferences for travel and BE are the primary sources of residential self-selection because people choose a travel mode may be caused by their preferences but not the BE. Future work should address the self-selection effects when exploring the relationships.

1.26 Pg. 12, line 369: Also, what about the focus on only commute mode? What other aspects of travel behavior would be important to study?

[Response] Thank you for comments. As you suggest, we have added a limitation as follows (lines 411-414): Third, this study only focuses on commuting mode choice. However, the neighborhood BE should be associated with other aspects of travel behavior. Thus, the associations between BE and other aspects of travel behavior need to be explored in future work.

1.27 Pg. 12, line 374: “provide appropriate implications” does not make sense. Studies do not “provide” implications. And what are “appropriate implications”?

[Response] Thank you for comments. As you suggest, we have deleted related contents.

1.28 Section 6 is a summary of findings, not really a conclusion.

[Response] Thank you for comments. As you suggest, we have revised the section of Conclusion. First, we summary the study using a sentence (lines 388-389). Second, we present the research contribution of this study (lines 389-391). Third, we present a summary of findings (lines 391-403). Finally, we give the limitations of this study (lines 404-414). The revised conclusion is as follows (lines 388-414): This study provides new insights into the effects of the BE variables on commuting mode choice based on a large-scale survey dataset from China. It contributes to the lit-erature by providing an analysis of the relationship between BE and commuting mode choice from both cross-sectional and longitudinal perspectives within a unified framework. The findings indicate that both life-cycle events and changes in BE play remarkable roles in affecting commuting mode shift. The effects of changes in BE sug-gest that promoting more balanced land use and improving residential density are important for reducing car use and encouraging active travel. Additionally, increases in land use mix precipitate the use of motorcycles and electric-bikes, but reduce the use of public transit. Changes in distance to the town government or neighborhood com-mittee are associated with most commuting mode shift, whereas changes in commu-nity types are only associated with switching to motorcycle/electric-bike. These find-ings largely support the cross-sectional results. Apart from changes in BE, most life events also have significant effects on commuting mode shift. The findings may not transfer directly to other countries owing to differences in land use and transport, and societal and institutional factors. However, it has important policy implications for other metropolitan regions in China.

Several limitations should be addressed. First, this study only captures direct rela-tionships among BE, life-cycle events and commuting mode shift. Future work should consider direct and indirect effects to capture the complex influence mechanism among them in a unified framework. Second, due to data unavailability, preferences for travel and BE are not considered. Preferences for travel and BE are the primary sources of residential self-selection because people choose a travel mode may be caused by their preferences but not the BE. Future work should address the self-selection ef-fects when exploring the relationships. Third, this study only focuses on commuting mode choice. However, the neighborhood BE should be associated with other aspects of travel behavior. Thus, the associations between BE and other aspects of travel be-havior need to be explored in future work.

Reviewer 2 Report

Dear authors!

I would like to congratulate the authors for their excellent work. They have done an excellent write-up with detailed discussion and motivations of this study.

And the results are very interesting. Maybe just a small question or remark to section Discussion or Conclusion: how the results could be used? Any strategy or policy or campaign behind it? Usually, we want to change travel habits in terms of inactive to active travel or from car to bicycle, or from transport with a higher carbon footprint to a lower one.

I suggest you change the little bit the title or answer to your question » Correlation or causality?« in Conclusion. Correlation is usually “proved” by a favorable correlation coefficient.

Regards.

Author Response

- Reviewer 2

I would like to congratulate the authors for their excellent work. They have done an excellent write-up with detailed discussion and motivations of this study.

[Response] Thank you for your comments. We have revised the related contents according to the requirements. We are hoping that the revised manuscript has properly responded to all your comments.

2.1 And the results are very interesting. Maybe just a small question or remark to section Discussion or Conclusion: how the results could be used? Any strategy or policy or campaign behind it? Usually, we want to change travel habits in terms of inactive to active travel or from car to bicycle, or from transport with a higher carbon footprint to a lower one.

[Response] Thank for your comments. As you suggest, we have supplied related policy implications in the section of Discussion.

About urban planning, we have supplied the following contents (lines 354-359): These findings demonstrate the effectiveness of these planning strategies. Additionally, urban sprawl motivated by urbanization has a negative effect on sustainable developments because people tend to commute by car, but have a lower probability of commuting by public transit in suburbs. To encourage public transit use in suburbs, it is of great importance to ensure timely provision of public transit, which has also been highlighted by many cross-sectional studies [11].

About transport policies, we have supplied the following contents (lines 379-384): For example, transport policies, such as subsidizing policies for motorcycles and electric-bikes, may be important to avoid excessive use of cars for young couples because marriage and childbirth can increase motorized travel demand. The subsidizing policies for attractive alternatives can be effective because an earlier study suggests that informing young people about the availability of public transit causes more sustainable travel [71].

2.2 I suggest you change the little bit the title or answer to your question » Correlation or causality?« in Conclusion. Correlation is usually “proved” by a favorable correlation coefficient.

[Response] Thank for your comments. As you suggest, we have revised the title as follows: Exploring the relationship between built environment and commuting mode choice: Longitudinal evidence from China

Reviewer 3 Report

This paper examines the relationships of the built environment (BE)  with commuting mode choice from both cross-sectional and causal perspectives. In addition, the effects of life-cycle events are considered in the commuting mode choice. Notably, the data from two rounds of the China Labor-force Dynamics Survey (CLDS) are employed which explores the change of the travel choice in developing countries. The results are clear and interesting. There are a certain of ambiguous parts in the current version of the manuscript. My comments are as below:

1)  The first contribution is not very clear. Why it is significant to examine relationships of commuting mode choice from both cross-sectional and causal perspectives. Some additional literature review may be required to support this contribution.

2) What type of BE change is considered in this study? Does the BE only refer to the area around the travelers' residence?

3) In line 170 page 4, what does the active mode refer to for commuting travel? The authors may need to give some examples.

4) Please provide some explanations about the procedure of "deliberation" in Figure 2.

5) As the definition of the distance to the local administrative center (DTAC), what does the "local administrative center" refer to here?

6) The logit regression model is used in this study. The authors may want to discuss why this model is chosen. Is it possible to use other models to conduct similar research?

Author Response

- Reviewer 3

This paper examines the relationships of the built environment (BE)  with commuting mode choice from both cross-sectional and causal perspectives. In addition, the effects of life-cycle events are considered in the commuting mode choice. Notably, the data from two rounds of the China Labor-force Dynamics Survey (CLDS) are employed which explores the change of the travel choice in developing countries. The results are clear and interesting. There are a certain of ambiguous parts in the current version of the manuscript. My comments are as below:

[Response] Thank you for your comments. We have revised the related contents according to the requirements. We are hoping that the revised manuscript has properly responded to all your comments.

3.1 The first contribution is not very clear. Why it is significant to examine relationships of commuting mode choice from both cross-sectional and causal perspectives. Some additional literature review may be required to support this contribution.

[Response] Thank for your comments. In order to make the contribution clearer, we have supplied some contents to describe the research gap that we want to fill as follows (lines 40-53): First, knowledge about BE and commuting mode choice are primarily analyzed based on cross-sectional data [7-9], and the longitudinal link between BE and commuting mode choice remains underexplored. Most cross-sectional studies have demonstrated that surrounding BE attributes are determinants of commuting mode choice [10, 11]. Cross-sectional studies cannot meet the time precedence [4, 7, 8], which is criteria for inferring causality. Thus, it is important to model BE and commuting mode choice based on longitudinal data. Some longitudinal studies have found that people usually change their travel behavior following a residential relocation [12-14]. The changes may be primarily explained by the reason that changes in BE and social environment change the choice sets or the utilities of the choices, thus triggering a reconsideration of people’s travel behavior [15]. Therefore, it is expected that BE has a longitudinal rela-tionship with commuting mode choice. In other words, changes in BE may precipitate commuting mode shift. However, it remains largely unclear whether and to what ex-tent BE changes precipitate changes in commuting mode choice.

Moreover, to make the first contribution clearer, we have also re-written the first contribution in the manuscript as follows (lines 74-77): This study provides additional evidence on the relationships between commuting mode choice and BE from a longitudinal perspective. We also perform comparative analysis of the cross-sectional and longitudinal relationships between BE and commuting mode choice.

3.2 What type of BE change is considered in this study? Does the BE only refer to the area around the travelers' residence?

[Response] Thank for your comments. As we described in the section of literature review, the BE at different spatial scales has effects on travel behavior. In this study, we only focus on the BE around residences. To make this clearer, we have supplied the following sentence in the new version (lines 222-223): Regarding BE variables and their changes, we only focus the BE around residential lo-cations.

3.3 In line 170 page 4, what does the active mode refer to for commuting travel? The authors may need to give some examples.

[Response] Thank for your comments. We have supplied related contents as follows: active modes (i.e., walking and cycling).

3.4 Please provide some explanations about the procedure of "deliberation" in Figure 2.

[Response] Thank for your comments. The deliberation is procedure is a decision procedure about people’s commuting behavior. People will decide whether they should change their commuting behavior because of changes in their life or contexts. As you suggest, we have supplied more explanations about the procedure as follows (lines 158-164): From the longitudinal perspective, most people deliberate their daily travel choices, which tend to be habitual, only when they experience a significant change in their life contexts [2]. Among contextual changes, life-cycle events and BE changes are essential triggers. People’s commuting behavior is not deliberated over each time. It is performed until life-cycle events disrupt everyday routines [1, 64, 65]. These events may trigger a reconsideration of their commuting behavior, and thus cause commuting mode shift.

3.5 As the definition of the distance to the local administrative center (DTAC), what does the "local administrative center" refer to here?

[Response] Thank for your comments. Local administrative center refers to the town government or neighborhood committee in the survey. If the resident is located in rural areas, it refers to the town government. If the resident is located in rural areas, it refers to the neighborhood committee. The two types of agencies are local administrative centers in rural and urban areas. As you suggest, we have revised it in the manuscript.

3.6 The logit regression model is used in this study. The authors may want to discuss why this model is chosen. Is it possible to use other models to conduct similar research?

[Response] Thank for your comments. As you suggest, we have supplied why this model is chosen in the new version as follows (lines 244-249): Commuting mode choice or shift is a typical discrete choice problem. People tend to choose the option with the maximum utility in the choice set. The utility function con-sists of a fixed and random part. In the cross-sectional model, the fixed part consists of socio-economics and neighborhood BE. In the longitudinal model, the fixed part con-sists of life-cycle events and changes in BE besides socio-economics and neighborhood BE. In logit models, the random part is assumed to follow a Gumbel distribution.

The probit model can also be used to solve the discrete choice problem and it is also one type of maximum utility model. We also conduct a probit model when we write this paper. But the logit model performs better than the probit model. So we choose the logit model in the paper.

Round 2

Reviewer 3 Report

Thanks to the authors for addressing all my questions. I have no further comment on the manuscript.